# Nonlinear temperature effects on multifractal complexity of metabolic rate of mice

Fabio A. Labra[1,2], Jose M. Bogdanovich[2,3] and Francisco Bozinovic[3]

[1] Facultad de Ciencias, Universidad Santo Tomás, Santiago, Chile
[2] Centro de Investigación e Innovación para el Cambio Climático, Universidad Santo Tomás, Santiago, Chile
[3] Departamento de Ecología, Center of Applied Ecology & Sustainability (CAPES) and LINC-Global, Facultad de Ciencias Biológicas, Pontificia Universidad Católica de Chile, Santiago, Chile

## ABSTRACT

Complex physiological dynamics have been argued to be a signature of healthy physiological function. Here we test whether the complexity of metabolic rate fluctuations in small endotherms decreases with lower environmental temperatures. To do so, we examine the multifractal temporal scaling properties of the rate of change in oxygen consumption $r(VO_2)$, in the laboratory mouse *Mus musculus*, assessing their long range correlation properties across seven different environmental temperatures, ranging from 0 °C to 30 °C. To do so, we applied multifractal detrended fluctuation analysis (MF-DFA), finding that $r(VO_2)$ fluctuations show two scaling regimes. For small time scales below the crossover time (approximately $10^2$ s), either monofractal or weak multifractal dynamics are observed depending on whether $T_a < 15$ °C or $T_a > 15$ °C respectively. For larger time scales, $r(VO_2)$ fluctuations are characterized by an asymptotic scaling exponent that indicates multifractal anti-persistent or uncorrelated dynamics. For both scaling regimes, a generalization of the multiplicative cascade model provides very good fits for the Renyi exponents $\tau(q)$, showing that the infinite number of exponents $h(q)$ can be described by only two independent parameters, $a$ and $b$. We also show that the long-range correlation structure of $r(VO_2)$ time series differs from randomly shuffled series, and may not be explained as an artifact of stochastic sampling of a linear frequency spectrum. These results show that metabolic rate dynamics in a well studied micro-endotherm are consistent with a highly non-linear feedback control system.

Corresponding author
Fabio A. Labra, flabra@santotomas.cl, fabio.labra@gmail.com

## INTRODUCTION

Physiologic complexity is ubiquitous in all living organisms (*Bassingthwaighte, Liebovitch & West, 1994*; *Glass, 2001*; *Goldberger et al., 2002*; *Burggren & Monticino, 2005*). It emerges as the result of interactions among multiple structural units and regulatory feedback loops, all of which function over a wide range of temporal and spatial scales, allowing the organism to respond to the stresses and challenges of everyday life (*Bassingthwaighte, Liebovitch & West, 1994*; *Goldberger et al., 2002*). As a consequence of these intricate

regulation feedbacks, most physiological state variables typically present non-linear non-stationary dynamics, with irregular fluctuations that follow power-law probability distributions and present long-range correlations over multiple time scales (*Glass, 2001*; *Goldberger & West, 1987*; *Kantelhardt, 2011*; *Labra, Marquet & Bozinovic, 2007*; *Mantegna & Stanley, 2000*; *Bassingthwaighte, Liebovitch & West, 1994*). The application of analytic techniques from nonlinear dynamics and statistical physics to the study of different physiologic variables has led to the proposition of a general theory to account for the complexity of physiologic variables (*Glass, 2001*; *Costa, Goldberger & Peng, 2002*; *Goldberger et al., 2002*; *Kantelhardt, 2011*; *Lipsitz, 2004*). This theory states that, given certain parameter conditions, the state variables of healthy systems reveal complex variability associated with long-range (fractal) correlations, along with distinct classes of nonlinear interactions (*Goldberger, 1996*; *Goldberger, Rigney & West, 1990*; *Goldberger et al., 2002*). Over the last two decades, different studies have shown that the break down of this type of multi-scale, nonlinear complexity is a characteristic signature of disease and senescence, and as a result, the study of complexity in physiological variables has shown important promise in the efforts to understand and diagnose different pathologies (*Costa et al., 2008*; *Delignières & Torre, 2009*; *Goldberger et al., 2002*; *Hausdorff et al., 2001*; *Hu et al., 2004*; *Ivanov et al., 2007*; *Lipsitz, 2004*).

While different quantitative approaches have been devised to measure the degree of complexity in physiological signals (e.g., *Burggren & Monticino, 2005*; *Costa, Goldberger & Peng, 2002*; *Feldman & Crutchfield, 1998*; *Pincus, 1991*; *Rezek & Roberts, 1998*; *Richman & Moorman, 2000*; *Schaefer et al., 2014*), most studies examining changes in physiological complexity as a result of pathological alterations have been conducted by examining either the change or loss of longrange correlations of physiologic signals (e.g., *Costa et al., 2008*; *Delignières & Torre, 2009*; *Goldberger et al., 2002*; *Hausdorff et al., 2001*; *Hu et al., 2004*; *Ivanov et al., 2007*; *Lipsitz, 2004*). Long-range correlated time series typically exhibit slowly decaying autocorrelation functions $C(s)$ across different time scales $s$, which are characterized by power law decay:

$$C(s) \propto s^{-\gamma} \tag{1}$$

with scaling exponent taking values in the range $0 < \gamma < 1$, such that a characteristic correlation time scale cannot be defined (*Chaui-Berlinck et al., 2002a*; *Chaui-Berlinck et al., 2002b*; *Billat et al., 2006*; *Kantelhardt, 2011*). It has been argued that the lack of a characteristic scale in physiological systems may help the organism to be more stable and adaptive to internal and external perturbations by preventing the emergence of periodic behaviors or phase locking, thus avoiding any restriction to the functional responsiveness of the organism in the face of external perturbations (*Peng et al., 1993*; *Peng et al., 2002*; *West & Shlesinger, 1989*). If this were correct, the study of long-range correlations would provide important insights on the degree of regulation and homeostasis of living organisms, as well as potential tools in the diagnosis of certain pathologies. A power law scaling of the spectrum of Fourier frequencies may also describe the presence of long-term correlations in any given stationary physiological signal:

$$S(f) \sim f^{-\beta}. \tag{2}$$

Long-range correlated processes of this type are often referred to as $1/f^{\beta}$ processes or noises, and are characterized by a unique value of the scaling exponent $\beta$, which provides a measure of the type of long-range correlation (*Chaui-Berlinck et al., 2002a*; *Chaui-Berlinck et al., 2002b*; *Billat et al., 2006*; *Kantelhardt, 2011*; *Schaefer et al., 2014*). Again, the power law scaling implies that no single characteristic scale may be identified. The Fourier power spectrum scaling exponent may be related to the correlation function exponent by the relationship $\beta = 1 - \gamma$. Further, the different scaling exponent values are associated with different types of correlation structure in a given time series or signal. Thus, for processes where $\beta = 0$ (or $\gamma = -1$) the signal shows no long-range correlation between values while values where $\beta > 0$ (or $\gamma > -1$) describe a process with long-range correlation or persistence. Processes where $\beta < 0$ (or $\gamma < -1$) describe a signal with long-range anti-correlations, or anti-persistence, where large values are followed by small ones (*Witt & Malamud, 2013*). Nevertheless, the use of frequency spectra requires not only that the time series be stationary, but also the use of particular binning procedures as well as averaging over a large number of realizations in order to accurately estimate the value of the scaling exponent $\beta$ (*Kantelhardt, 2011*; *Witt & Malamud, 2013*). An alternative approach for non-stationary time series is to characterize its long-range persistence by examining the self-affinity of the profile or cumulative sum $z_i = \sum r(VO_2, i)$, for all samples $i = 1$ to $N$ (*Peng et al., 2002*; *Kantelhardt, 2011*). Examination of these time series requires us to take into account that the time axis and the axis of the measured values $x(t)$ are not equivalent quantities, and that a rescaling of time $t$ by a factor $a$ may require rescaling of the series values $x(t)$ by a different factor $a^H$ in order to obtain a signal that is statistically self-similar to the original one (*Kantelhardt, 2011*). Hence, the exact type of self-affinity or statistical self-similarity in a time series may be described by the resulting scaling relation $x(t) \to a^H x(at)$ where $H$ corresponds to the Hurst exponent, which measures the degree of persistence or predictability of the profile or cumulated time series (*Kantelhardt, 2011*). The exponent $H$ may be studied by different methods including rescaled range analysis, fluctuation analysis, and detrended fluctuation analysis (*Peng et al., 2002*; *Kantelhardt, 2011*). In particular, Detrended fluctuation analysis (DFA) has been widely employed to reliably detect long-range autocorrelations in non-stationary time series, with a large number of studies using it to report long-range autocorrelations, although a few studies have reported anti-persistent anti correlations (e.g., *Bahar et al., 2001*; *Delignières et al., 2006*; *Delignières, Torre & Bernard, 2011*; *Kantelhardt, 2011*). The value of the Hurst exponent $H$ may be approximated by the DFA, which calculates the scaling of mean-square fluctuations with time series scale, yielding the scaling exponent $\alpha$ (*Feder, 1988*; *Hurst, 1951*; *Peng et al., 2002*; *Kantelhardt, 2011*). When DFA scaling relationships are observed, the scaling exponent $\alpha \approx H$ is related to the correlation exponent $\gamma$ by the relationship $\alpha = 1 - \gamma/2$, with $\alpha = 0.5$ being the threshold between anti persistence and persistence (*Peng et al., 2002*; *Kantelhardt, 2011*).

Despite the increased interest to study fractal or long-range correlated dynamics across many systems, in some highly nonlinear complex systems, the resulting time series presents a scaling autocorrelation function and frequency power spectrum which may be better described by a large number of scaling exponents rather than by a single scaling exponent value (*Kantelhardt, 2011*). Thus, one may distinguish between monofractal and multifractal

 

signals. Monofractal signals present a long-range correlation structure where a single scaling exponent suffices to describe the correlation scaling. On the other hand, multifractal signals require an infinite spectrum of scaling exponents to describe their correlation structure (*Humeau et al., 2009*; *Ivanov et al., 1999*; *Kantelhardt, 2011*; *Suki et al., 2003*; *West & Scafetta, 2003*). Thus, multifractal time series are heterogeneous, showing a given value of the self-affinity exponent only in local ranges of the signal structure, such that their self-affinity exponent varies in time. Hence, multifractal signals may be characterized by a set of local fractal sets that represent the support for each Hurst exponent value (*Bassingthwaighte & Raymond, 1994*; *Ivanov et al., 1999*; *Kantelhardt, 2011*). In this regard, multifractal time series are more complex than monofractal ones, and determining whether a given complex physiologic system presents monofractal or multifractal dynamics may provide insight on the degree of complexity or nonlinearity of the underlying control mechanisms (*Mantegna & Stanley, 1997*).

In endotherms, metabolic rate ($VO_2$) is a global emergent property that reflects the sum of the energetic costs required to maintain homeostasis allowing body temperature ($Tb$) to remain as constant as possible despite any changes of its surrounding ambient temperatures ($Ta$) (*Karasov & Martinez del Rio, 2007*; *Lighton, 2008*; *McNab, 2002*). Under controlled laboratory conditions, it is possible to identify a range of optimal $Ta$ values where $Tb$ may be kept constant without changes in energy expenditure, but rather as a result of adjustments to physical processes (i.e., conductance, radiation, and convection). Within this range of $Ta$ values $VO_2$ is expected to show minimal variation, and hence it is named the thermoneutral zone ($TNZ$) (*Bozinovic & Rosenmann, 1988*; *Chaui-Berlinck et al., 2005*; *Karasov & Martinez del Rio, 2007*; *Lighton, 2008*; *Lipsitz, 2004*; *McNab, 2002*). A striking characteristic of $VO_2$ signals is that, even within the $TNZ$ they may be non-stationary, showing changes in the mean and variance of the time series (*Chaui-Berlinck et al., 2002a*). Studies with small endotherms have shown that $VO_2$ dynamics within the $TNZ$ present irregular fluctuations with long-range correlations, evidenced by the presence of a single monofractal $1/f^{\beta}$ scaling exponent in the Fourier frequency spectrum (*Chaui-Berlinck et al., 2002a*; *Chaui-Berlinck et al., 2002b*; *Billat et al., 2006*). Thus, within the $TNZ$, $VO_2$ shows complex dynamics that are consistent with a dynamical system under non-linear control (*Chaui-Berlinck et al., 2005*). The non-stationary behaviour in metabolic rate may be examined by analysing the rate of change in oxygen consumption, $r(VO_2)$ as a measure of the fluctuations of $VO_2$. It is defined as $r(VO_2) = \log10[VO_2(t+1)/VO_2(t)]$ (*Labra, Marquet & Bozinovic, 2007*). This variable reveals whether clusters of large, abrupt changes may be seen in the $r(VO_2)$ time series, or if similar variability is observed throughout. In addition, the calculation of $r(VO_2)$ allows the de-trending of the data, yielding a much more stationary time series. Examination of $r(VO_2)$ time series for different species of small mammals, birds and reptiles have shown that this variable has a symmetric power law probability distribution, centered in $r(VO_2) = 0$, with a universal triangular shape that does not change across different species (*Labra, Marquet & Bozinovic, 2007*). Thus, metabolic rate fluctuations follow a single statistical distribution despite differences in cardiovascular and respiratory designs, with distribution width scaling inversely with individual body size (*Labra, Marquet & Bozinovic, 2007*). However, to date, the correlation

structure in $r(VO_2)$ has not been examined. In a similar fashion to other complex non-linear time series, long-term correlations in $r(VO_2)$ would mean that large fluctuations are more likely to be followed by another large oscillation, while a small oscillation is likely to be followed by a small oscillation (*Ashkenazy et al., 2003*; *Bunde & Lennartz, 2012*). If this were the case, the expected average value of $VO_2$ would increase, showing a persistent trend. For $VO_2$ to show homeostatic regulation; however, its fluctuations would be expected to show anti-persistence over at least at some scales, so that large $r(VO_2)$ increases may be followed by large $r(VO_2)$ decreases, ensuring that overall average $VO_2$ values remain under homeostatic control. Thus, the presence of anti-persistent correlations may be expected for $r(VO_2)$ time series, particularly if there are strong control feedback loops regulating total energy expenditure in an organism. This suggests that examination of the type of autocorrelations present in $r(VO_2)$ time series, as well as the range of time scales involved may provide insight on the regulation feedback that may be acting on metabolic rate at the level of the organism. To gain some understanding of how this may be so, we examine the relationship between thermal stress and $VO_2$ fluctuations.

In endotherms, $VO_2$ fluctuations are expected to be proportional to the environmental thermal challenges measured as changes in the difference $(Tb - Ta)$ (*Bozinovic & Rosenmann, 1988*; *Chaui-Berlinck et al., 2005*; *Karasov & Martinez del Rio, 2007*; *Lighton, 2008*). Outside the *TNZ*, adjustments to the body's thermal conductance are not enough to sustain thermal homeostasis, and consequently additional physiological and biochemical process are required in order to keep constant the internal state, which leads to an increase both $VO_2$ and presumably $r(VO_2)$ as well. In the case of small endotherms, their body size leads to higher challenges associated to the loss of temperature resulting from the large body surface through radiation (*Chaui-Berlinck et al., 2005*; *Karasov & Martinez del Rio, 2007*; *Lighton, 2008*; *Lipsitz, 2004*; *McNab, 2002*). Given the intricate nature of the network of control processes involved in achieving constant *Tb* (*Chaui-Berlinck et al., 2005*), it is reasonable to expect that when faced with lower environmental temperatures values below the *TNZ* endothermic homeostatic processes would be accompanied by a more complex pattern of auto-correlations. To determine whether this is the case, we use fractal and multifractal analysis to examine whether the correlation structure of $VO_2$ shows any changes as a result of decreasing environmental temperatures In this regard, a working hypothesis is that for *Ta* values below the *TNZ* the $r(VO_2)$ signal should show a more complex pattern of long-range correlations, resulting in a broader range of autocorrelation scaling exponents, as expected for multifractal signals. These changes should come about as a result of the activation of internal feedback mechanisms to regulate *Tb*. A related question to this prediction concerns the form of this possible relation between complexity and decreasing of *Ta*. Records in wild rodents show a monotonic and linear increment of average $VO_2$ in animals exposed to *Ta* decreasing (30 °C–0 °C) (*Bozinovic & Rosenmann, 1988*), suggesting that $VO_2$ and $r(VO_2)$ complexity levels may also increase linearly. An alternative outcome may be the gradual decrease and eventual loss of complexity, due to a drop in the efficiency of the thermoregulatory feedback control at lower temperatures (*Angilletta, 2006*; *McNab, 2002*). This second pattern would be in agreement with the hypothesis of loss of physiological complexity in the face of extreme system degradation or acute

stress (*Goldberger et al., 2002*). To test these hypotheses, we examine the fractal properties of time series of $r(VO_2)$ measurements n laboratory mice (*Mus musculus*) exposed to environmental temperatures ranging from *TNZ* (30 °C in this species) to 0 °C. Thus, as first step in this work we assess whether $r(VO_2)$ values exhibit either monofractal or multifractal long-term correlations under different environmental temperatures. We do this by testing whether metabolic rate fluctuations show any longrange correlations and, if so, testing whether there may be described either by a single scaling exponent or if multiple scaling exponents are required, using the multifractal detrended fluctuation analysis (MF-DFA) method. We then assess how these quantitative descriptors of longrange correlations vary with environmental temperature, assessing how they change with decreasing values of *Ta*.

## METHODS

### Determination of metabolic rate

Empirical $VO_2$ time series were determined by measuring metabolic rate in wild-type male white laboratory mice. Mice were transferred to the laboratory and housed individually with sawdust bedding. Mice were provided with water and fed with food pellets *ad libitum*. Ambient temperature and photoperiod were held constant at $20 \pm 2$ °C and 12L:12D respectively. Care of experimental animals was in accordance with institutional guidelines. The Bioethics commissions of Universidad Santo Tomás, Pontificia Universidad Católica de Chile, and The Chilean National Committee of Science and Technology (CONICYT) approved all experimental protocols followed. Animals were held under these conditions for two weeks prior to measurements and then fasted for 3 h immediately prior to metabolic rate records in metabolic chambers (*Lighton, 2008*). Individuals were measured at seven different *Ta*, 0 °C, 5 °C, 10 °C, 15 °C, 20 °C, 25 °C and 30 °C, with the latter corresponding to the lower limit of *TNZ* in this species. Overall, 18 individuals were assigned to different temperature treatments, with the order of temperature treatments for each individual assigned at random to avoid any artefacts. In addition, colonic body temperature ($T_b$) was recorded at the end of each measurement using a Digi-Sense copper-constant thermo-couple to evaluate a possible torpor condition at the end of the experiment. In each experimental record $VO_2$ was measured in a computerized open-flow respirometry system (Sable Systems, Las Vegas, Nevada). The metabolic chamber received dried air at a rate of 800 ml/min from mass flow-controllers (Sierra Instruments[TM], Monterey, California), which ensured adequate mixing in the chamber. Air passed through $CO_2$ and $H_2O$ absorbent granules of Baralyme[TM] and Drierite[TM] respectively before and after passing through the chamber and was monitored every 1 s. This allowed us to obtain time series of oxygen consumption recorded at periodic intervals of $t = 1$ s. After the $r(VO_2)$ time series were registered, they were then analysed by calculating the corresponding $r(VO_2)$ time series.

### Assessing long range correlations in metabolic rate

To determine the presence of long-term correlations in the $r(VO_2)$ time series, we examined the power spectral density $S(f) \equiv |x(f)|^2$, where $x(f)$ is the Fourier transform of $r(VO_2)$ data observations measured under experimental conditions ($x_i$) evaluated at frequencies $f = 0, \ldots, N/2$ (*Bunde & Lennartz, 2012*; *Kantelhardt, 2011*). As mentioned above, for
long-term correlated time series, it can be shown that the power spectral density decays with frequency following a power law (see Eq. (2)). In order to avoid potential artefacts due to lack of stationary behaviour, we also used the Detrended Fluctuation Analysis method (DFA) (*Kantelhardt, 2011*; *Peng et al., 1995a*). Briefly, DFA analyses a profile or accumulated data series $z_i = \sum r(VO_2, i)$, for all samples $i = 1$ to $N$. The profile is divided into $N_s$ non-overlapping segments of scale s. For every segment $v$, the local trend is fit by a polynomial of order $n$, and the variance raised to the 2th power $[\sigma^2(v,s)]^2$ between the local trend and the profile in each segment $v$ is calculated. The mean fluctuation function $F(s)^2$ is then calculated by:

$$F_2(s) \equiv \left\{ \frac{1}{N_s} \sum_{v=1}^{Ns} [s^2(s)] \right\}^{1/2}. \tag{3}$$

Examination of how $F_2(s)$ scales with box size or scale $s$ allows the estimation of the scaling exponent $\alpha_{DFA}$, which is often referred to as the global Hurst exponent $H$ (*Goldberger et al., 2002*; *Ivanov et al., 2007*; *Kantelhardt, 2011*; *Peng et al., 1995a*). When observed time series are either uncorrelated or show short term correlations, $\alpha_{DFA} = 0.5$ (*Kantelhardt, 2011*; *Peng et al., 1995a*). For long-term correlated data with persistent $1/f^\beta$ noise, where $\beta = 1.0$, $\alpha_{DFA}$ exhibits values of equal to 1.0. For values of $\alpha_{DFA}$ below 0.5, the series is said to be anti-persistent, with positive trends being associated with negative trends (*Delignières et al., 2006*; *Delignières, Torre & Bernard, 2011*).

## Assessing multifractality of metabolic rate

To determine the presence of multifractality in the fluctuations of metabolic rate we applied multifractal detrended fluctuation analysis (MF-DFA) (*Kantelhardt, 2011*; *Kantelhardt et al., 2002*) to $r(VO_2)$ data measured under experimental conditions. This method yields similar results to other existing methods of multifractal analysis in time series (*Ivanov et al., 2007*; *Kantelhardt, 2011*; *Kantelhardt et al., 2002*; *Ludescher et al., 2011*; *Oswiecimka, Kwapien & Drozdz, 2006*), but is considerably easier to implement, being based on an extension of DFA (*Kantelhardt, 2011*; *Kantelhardt et al., 2002*; *Ludescher et al., 2011*). Briefly, MF-DFA analyses a profile or accumulated data series $z_i = \sum r(VO_2, i)$, for all samples $i = 1$ to $N$. The profile is divided into $N_s$ non-overlapping segments of scale s. For every segment $v$, the local trend is fit by a polynomial of a given order $o$, where $o = 1$, 2 or 3. The resulting variance is then raised to the q/2th power $[\sigma^2(v,s)]^{q/2}$ between the local trend and the profile in each segment $v$ is calculated. When $q = 0$, logarithmic averaging may be applied (*Kantelhardt, 2011*; *Kantelhardt et al., 2002*; *Ludescher et al., 2011*). A generalized fluctuation function $F_q(s)$ is then calculated by averaging all the variances across all segments of scale $s$:

$$F_q(s) \equiv \left\{ \frac{1}{N_s} \sum_{v=1}^{Ns} [\sigma^2(v,s)]^{q/2} \right\}^{1/q}. \tag{4}$$

In general, $F_q(s)$ exhibits a scaling relationship with time scale $s$: $F_q(s) \sim s^{h(q)}$, which allows the estimation of a set of exponents $h(q)$ for every moment $q$. These scaling

exponents correspond to the generalized Hurst exponents. In some nonlinear complex systems, the $F_q(s)$ function has been shown to exhibit scaling crossovers, with more than one asymptotic scaling exponent (*Koscielny-Bunde et al., 2006*). Hence, we tested whether linear or piecewise linear regressions best fit the scaling relationship of $F_q(s)$ with $s$, using log-transformed data. The piecewise or segmented relationship between the mean response $\mu = E[Y]$ and the variable X, for observation $i = 1, 2, \ldots, n$ was modeled by adding the following terms in the linear predictor:

$$\beta + \beta_1 X_i + \beta_2 (X_i - \delta)+ \tag{5}$$

where $(X_i - g\delta)+ = (X_i - g\delta) \times I(X_i > \psi)$ and $\delta$ is the fitted breakpoint or crossover point and $I(\cdot)$ is an indicator function that is equal to one when the statement is true and is equal to zero when the statement is false (*Muggeo, 2003*). Piecewise linear models were fitted using the *segmented* library (*Muggeo, 2003*) in the R program (*R Development Core Team, 2014*). If no crossovers were observed, then linear regression would be favored over a piecewise regression. To test this, the *segmented* library uses Davie's test to test for a non-constant regression parameter in the linear predictor (*Muggeo, 2003*). Once the correct regression model is identified, the regression slopes provide the asymptotic estimates for the scaling exponents $h(q)$. If no crossover is present, only one scaling exponent $h(q)$ is obtained for every moment $q$. If a crossover point is detected, then two scaling exponents $h(q)$ and $h(q)$ are obtained for every moment $q$.

For monofractal self-affine time series, $h(q)$ is independent of the chosen moment $q$, and is identical to the global Hurst exponent $H$ regardless of the value of the moment $q$ (*Feder, 1988*; *Hurst, 1951*; *Kantelhardt et al., 2003*; *Kantelhardt et al., 2002*). Hence, for monofractal self-affine time series $\alpha_{DFA} \approx H$. On the other hand, in multifractal time series $h(q)$ varies with $q$, reflecting the fact that small and large fluctuations scale differently (*Kantelhardt et al., 2002*). For negative values of $q$, $h(q)$ describes the scaling behaviour of those time series segments with small fluctuations, whereas for positive values of $q$, $h(q)$ describes the scaling behaviour of those time series segments with large fluctuations (*Kantelhardt et al., 2002*). It has been shown that the generalized Hurst exponent $h(q)$ can be directly related to the classical multifractal scaling Renyi exponents $\tau(q)$ defined by the standard partition function-based formalism using the relationships: $\tau(q) = qh(q) - 1$ and $h(q) = (\tau(q) + 1)/q$ (*Kantelhardt et al., 2002*; *Koscielny-Bunde et al., 2006*). Thus, it may be shown for normalized, stationary time series that the multifractal spectra estimated by MF-DFA have a deep similarity with thermodynamics (*Kantelhardt et al., 2002*).

For monofractal records, $\tau(q)$ is a linear function of $q$, while multifractal records are characterized by non-linear dependence of $\tau(q)$ on $q$ (*Ivanov et al., 1999*; *Kantelhardt et al., 2002*; *Koscielny-Bunde et al., 2006*). Also, it can be shown that $h(q)$ may be related to the singularity spectrum $f(\alpha)$ via a Legendre transform:

$$f(\alpha) = q[\alpha - h(q)] + 1 \tag{6}$$

where $\alpha = [d\tau(q)/dq]$ is the singularity strength, or Hölder exponent, while $f(\alpha)$ denotes the singularity dimension of the subset of the time series that is characterized by a given

value of singularity strength $\alpha$ (*Feder, 1988*; *Kantelhardt et al., 2002*; *Ludescher et al., 2011*; *Ihlen, 2012*). For monofractal self affine signals, the singularity spectrum of the time series is a single point, showing that there is a unique value or a very small set of values of singularity strength $\alpha$, with a corresponding fractal dimension $f(\alpha) = 1$. For multifractal self affine signals, the singularity spectrum of the time series is a parabola, with a maximum at the dominant singularity strength observed in the time series.

To assess multifractality in $r(VO_2)$ time series, we calculated the fluctuation function $F_q(s)$ for data obtained from wild-type white laboratory mice $r(VO_2)$ time series measured under controlled conditions. Following recent studies, we fit both the $h(q)$ and $\tau(q)$ spectra with a modified version of the multiplicative cascade model, which has been proposed by (*Koscielny-Bunde et al., 2006*):

$$h(q) = (1/q) - (ln(a^q + b^q))/(q\,ln(2)) \tag{7}$$

and

$$\tau(q) = -(ln(a^q + b^q))/(ln(2)). \tag{8}$$

The modified multiplicative cascade model functions (MMCM) allows the description of multifractal spectra with only two parameters, $a$ and $b$, which take values between 0 and 1 with $a + b \geq 1$. An additional advantage is that these functions also extend to negative $q$ values, and thus allow estimation of the multifractal spectrum $f(\alpha)$ for these values as well (*Koscielny-Bunde et al., 2006*). Using the $\tau(q)$ spectra, we estimated the parameters $a$ and $b$ for Eq. (8), allowing us to obtain continuous $\tau(q)$ and $f(\alpha)$ spectra from the MMCM fits.

To test whether observed long term correlation behaviour was different from a random expectation, we randomized all time series using an amplitude-adjusted Fourier transform algorithm (AAFT) (*Schreiber & Schmitz, 1996*; *Schreiber & Schmitz, 2000*). The scaling functions were calculated for all surrogate time series and the corresponding scaling exponents (e.g., $\beta$ and $\alpha_{DFA}$ for Fourier spectral density and DFA respectively) were calculated (*Schreiber & Schmitz, 1996*; *Schreiber & Schmitz, 2000*).

## Assessing the effect of temperature on multifractality of metabolic rate fluctuations

As explained above, regular $VO_2$ time series were obtained under temperature-controlled conditions (see 'Methods' sections for details). To assess the effect of $Ta$ on long range and multifractal measures of $r(VO_2)$ fluctuations, we calculated the average fluctuation function $F_q(s)$ for each of the seven temperature treatment groups, testing whether the resulting $h(q)$ and $\tau(q)$ spectra are also multifractal. In order to summarize the observed results, we calculated the singularity spectrum $f(\alpha)$, which allows a compact description of the degree of multifractality through the quantification of $\Delta\alpha$, the width of the singularity spectrum as well as the average dominant exponent $\alpha_{max}$, which indicates which is the dominant scaling exponent, or the one which shows greater support on average across the time series. We then summarized the various spectra across the experimental temperature treatments, allowing us to examine their response to temperature. To test whether observed

multifractal behaviour was different from a random expectation, we randomized all time series using an amplitude-adjusted Fourier transform algorithm (AAFT) (*Schreiber & Schmitz, 1996*; *Schreiber & Schmitz, 2000*). After the surrogates were generated, the general fluctuation function $F_q(s)$ and the $h(q)$ spectra were calculated as explained above. We then compared $h(q)$, $\tau(q)$ and $f(\alpha)$ spectra for the shuffled time series. Again, we summarized the various spectra for shuffled time series across the experimental temperature treatments, allowing us to compare them with original time series spectra as for different temperature treatments. To assess the potential effect of de-trending polynomial order $o$, all data analyses were carried out for each individual time series were carried out using three orders: $o = 1, 2$ or 3. Data analyses were carried out using Matlab R2011b and R software (*R Development Core Team, 2014*).

## RESULTS

As described in the physiological literature for endotherms, average $VO_2$ values in the lab mouse show a marked thermal response below *TNZ*, with higher $VO_2$ values that increase away from basal metabolic rate (*BMR)* as *Ta* becomes progressively lower (Fig. 1A). None of the animals studied showed signs of torpor either during or after the $VO_2$ measurements, and observed *Tb* varied from 36.0 to 37.3 °C across all records. However, even within the *TNZ* (30 °C), typical $VO_2$ time series exhibit irregular non-stationary fluctuations (Fig. 1B). The rate of change $r(VO_2)$ yields a de-trended time series, which reveals abrupt changes in $VO_2$, with clusters of large fluctuations separated from clusters of smaller fluctuations(Fig. 1C). This suggests the presence of long-term correlation or persistence in these time series. The clustering of large fluctuations is lost when data are shuffled randomly using AAFT (Fig. 1D), providing indication that the observed pattern of $r(VO_2)$ fluctuations may be associated with the autocorrelation structure of the time series (*Schreiber & Schmitz, 1996*; *Schreiber & Schmitz, 2000*; *Kantelhardt, 2011*) rather than with the fat tailed probability distribution shown by this variable (*Labra, Marquet & Bozinovic, 2007*). The statistical pattern of autocorrelation in the sequence of large and small fluctuations may be examined by calculating the Fourier frequency power spectra, which reveals the presence of long-term correlations, shown by a 1/f-like scaling exponent (Fig. 1E). On the other hand, shuffled time series exhibit a shallower power spectrum, indicating the loss of these long-term correlations (Fig. 1E) (*Kantelhardt et al., 2002*; *Schreiber & Schmitz, 1996*; *Schreiber & Schmitz, 2000*). However, while $r(VO_2)$ time series do not exhibit obvious trends in the mean, they do show changes in variability through time, and as a result may not meet the statistical assumptions of spectral frequency estimation (*Kantelhardt, 2011*). Examination of detrended fluctuation analysis reveals a scaling crossover, with two clear scaling regimes shown by the root mean square fluctuation function $F_2(s)$ (Fig. 1F). This suggests that a single scaling exponent may not be sufficient to characterize the autocorrelation of $r(VO_2)$ fluctuations (*Kantelhardt et al., 2002*). In this time series, the scaling exponent for small time scales ($s < 100$ s), $\alpha_{DFA1}$, indicates the presence of persistent, long-range correlated fluctuations ($\alpha_{DFA1} = 0.91$) (Fig. 1F). However, for larger time scales ($s > 100$ s) we see that fluctuations over these time scales are

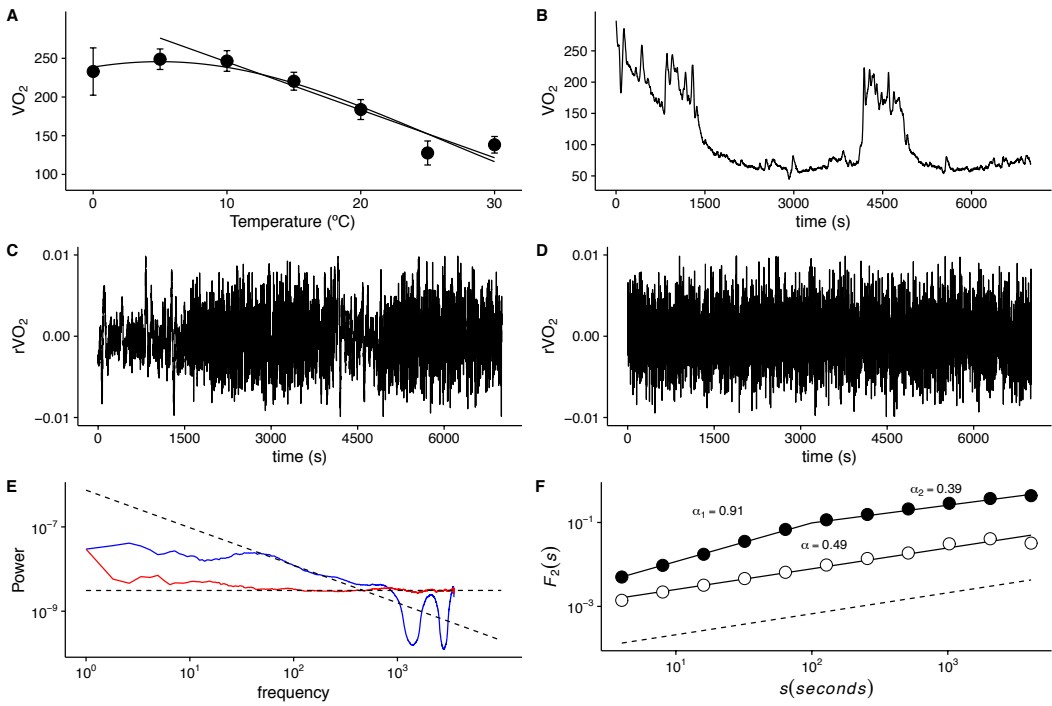

**Figure 1  Long-term correlations of metabolic rate fluctuations in *Mus musculus*.** (A) Average metabolic rates ($VO_2$) measured at different ambient temperatures. Average values $\pm$ standard errors are shown with open circles and error bars. Straight line shows calculated thermal conductance, while the humped curve corresponds to a fitted three parameter Gaussian function ($g(x) = a * \exp(-.5 * ((x - x0)/b)^2)$). (B) Metabolic rate ($VO_2$) time series shown for a representative individual measured at 30 °C for 1 3/4 h at 1 (s) intervals. Note the irregular, nonstationary dynamics, despite thermo neutral ambient temperature. (C) Observed $VO_2$ fluctuations $r(VO2) = \log10[VO_2(t+1)/VO_2(t)]$ time series for data in (B). Note the clustering of broad and narrow fluctuations. (D) Randomized $r(VO_2)$ values, showing the loss of the clustering of fluctuations. (E) Fourier power spectra for time series in (C) and (D) shown by blue and red lines respectively. A smoothing procedure was applied, which consisted of averaging the spectra for consecutive overlapping segments of 256 data points. Fitted OLS scaling relationships are shown in dotted lines. (F) Detrended fluctuation analyses (DFA) for the two time series shown in (C) and (D). Fluctuation functions for original and shuffled time series in are shown in open and filled circles respectively. Fitted scaling relationships are shown in dashed lines. Note the change in exponent values above $s = 100$ for the original time series.

anti-persistent, with the second scaling exponent $\alpha_{DFA2} = 0.39$ (*Eke et al., 2000*; *Delignières et al., 2006*; *Delignières, Torre & Bernard, 2011*). As mentioned above, in anti-persistent time series dynamics positive trends are usually followed by negative trends, thus showing a phenomenological signature of control or negative feedback over the rate of change of $VO_2$ (*Delignières, Torre & Bernard, 2011*). Shuffling the data results in a loss of the observed crossover scaling behaviour, indicating this is property is not a result of randomness in the pattern of fluctuations (Fig. 1F). Thus, we find that $r(VO_2)$ fluctuations within the TNZ show non-trivial long-range correlations, in agreement with previous observations for $VO_2$ in small endotherms (*Chaui-Berlinck et al., 2002a*; *Chaui-Berlinck et al., 2002b*). However, a single scaling exponent does not suffice to describe these long-range correlations.

When we examined the DFA scaling functions for $r(VO_2)$ fluctuations both within and outside the *TNZ*, we observe a similar crossover pattern across different temperatures, with
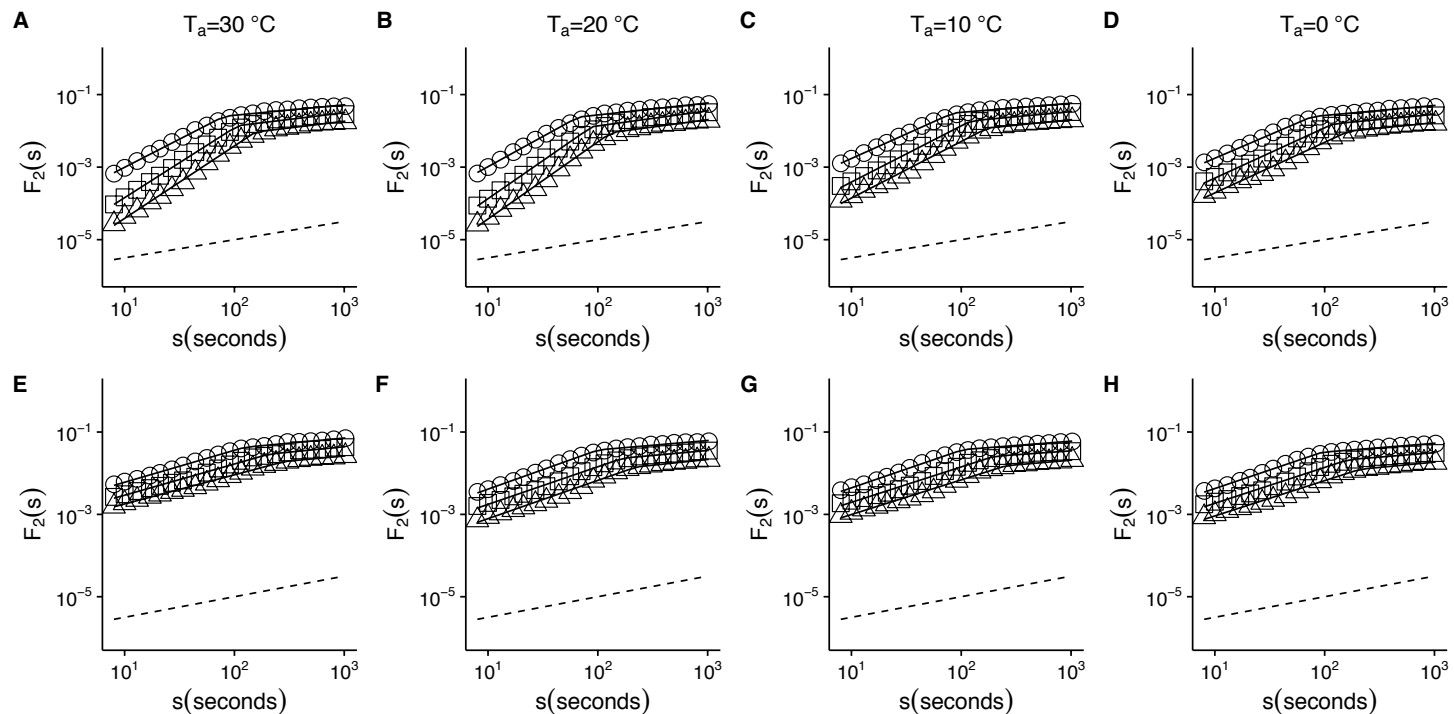

**Figure 2   Temperature effects on root-mean-square fluctuation function of $r(VO_2)$ in mice.** The figure shows the average $F_2(s)$ functions calculated with linear detrending for all mice. Results for the time series studied at 30 °C, 20 °C, 10 °C and 0 °C are shown in the respective columns arranged from left to right. Figures (A)–(D) show the average DFA functions calculated for the $r(VO_2)$ time series, while figures (E)–(H) show average DFA functions calculated for the AAFT shuffled data. All figures show the DFA root-mean-square fluctuation functions obtained using three different orders of detrending polynomials: linear (open circles), quadratic (open squares) and cubic functions (open triangles). Two scaling regimes can be observed across all temperatures and for all polynomial detrending orders. The first scaling regime spans scales between 8 and 100 s, while the second one spans scales from 100 to 1,024 s. All curves have been shifted vertically for clarity. Please note that while only four experimental temperatures are shown, the remaining three temperatures show similar patterns.

average $F_2(s)$ scaling functions show a crossover pattern which is similar to that observed in Fig. 1F. Hence, observed scaling exponent values for small to intermediate time scales) are consistent with persistent long-range autocorrelations (i.e., $0.5 < \alpha_{DFA1} < 1.0$) (Figs. 2A–2D). On the other hand, for intermediate to large scales, the scaling exponent values are consistent with anti-persistent long-range correlations ($\alpha_{DFA2} < 0.5$) (Figs. 2A–2D). Shuffling the individual time series results in changes to the $F_2(s)$ scaling functions, with average $\alpha_{DFA1}$ values becoming smaller (Figs. 2E–2H). Examination of the scaling exponent values shows that $\alpha_{DFA2}$ values do not show large changes for shuffled data (Fig. 3). This pattern is observed for linear (Fig. 3) as well as for quadratic and cubic de-trending orders $o$ (see Fig. S1). The existence of two scaling regimes for the long-range correlations of $r(VO_2)$ may be interpreted as evidence that two dominant scaling exponents may suffice to account for the correlation structure of the $r(VO_2)$ time series. An alternative possibility may be that a continuous spectrum of scaling exponents are required in order to account for the observed pattern of long-term correlations in $VO_2$ fluctuations. If the latter were the case, local scaling exponents would show a large number of possible values.

To visualize whether a sample $r(VO_2)$ time series is consistent with a multifractal process, we examined the changes in the value of local DFA scaling exponent $\alpha_{DFA}$ through time in

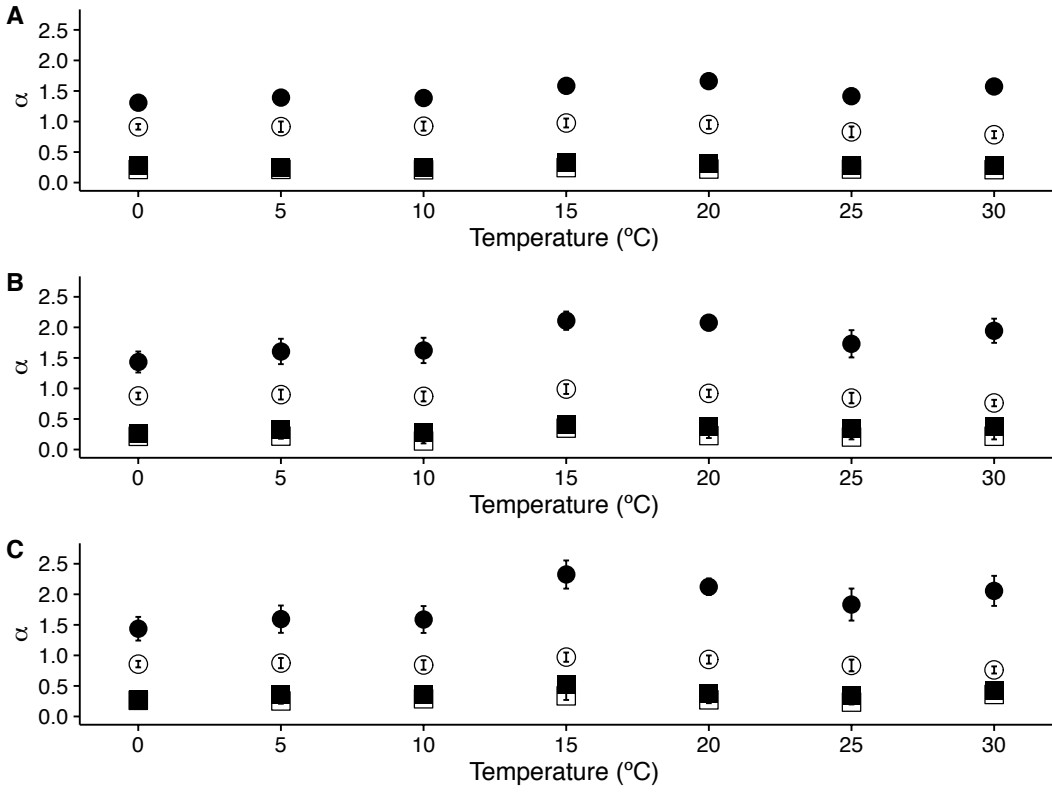

**Figure 3  Temperature effects on long range scaling exponent α in metabolic rate fluctuations.** The figure shows the average DFA scaling exponent $\alpha_{DFA}$ calculated as a function of experimental temperature. Average scaling exponents corresponding to exponent for raw $r(VO2)$ data within the $10 < s < 100$ scaling regime are shown with filled circles, while filled squares show the scaling exponents for the raw $r(VO2)$ data within the $100 < s < 1{,}024$ scaling regimes. Open circles and squares show the scaling exponents for these two respective scaling regimes when data are shuffled.

the time series shown in Fig. 1 (which was measured within the TNZ). We calculated the local value of $\alpha_{DFA}$ as for a moving window placed along the time series. We calculated $\alpha_{DFA}$ values using moving windows of 128, 256 and 512 s (Figs. 4A, 4B and 4C, respectively). All these window sizes correspond to the asymptotic exponent expected for the second scaling regime identified before for this time series (Fig. 1F). Observed local $\alpha_{DFA}$ exponent values change through time for all window sizes used, forming an irregular pattern (Fig. 4). Further, $\alpha_{DFA}$ values range broadly between 0.5 and 1.5, as shown by the blue lines in Fig. 4. Thus, while in some sections show exponent values close to 1.0, corresponding to persistent power law long-range correlations, other sections may show values closer to either 1.5 (corresponding to persistent Brownian motion) or to 0.5 (corresponding to uncorrelated fluctuations) (*Peng et al., 1995b*). There are also sections where the local $\alpha_{DFA}$ scaling exponent may take values below 0.5, corresponding to anti-persistent fluctuations (*Eke et al., 2000*; *Delignières et al., 2006*; *Delignières, Torre & Bernard, 2011*). Again, random shuffling of the time series destroys the observed pattern of irregular fluctuations of $\alpha_{DFA}$, with all exponent values clustering around 0.5, as shown by the red lines in Fig. 4. Thus,

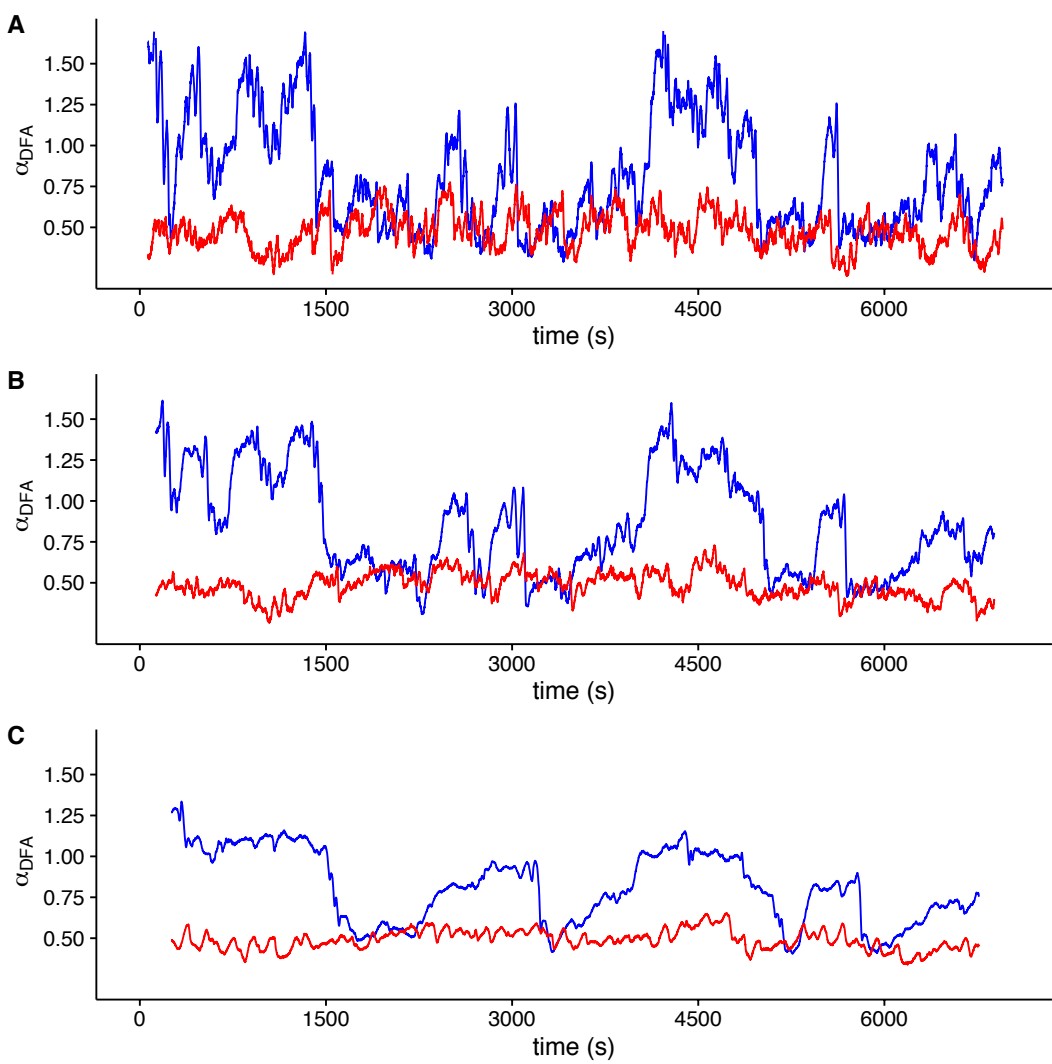

**Figure 4 Local DFA scaling exponents.** The figure shows the value of local DFA scaling exponents $\alpha_{DFA}$ for the time series in Figs. 1C (blue lines) and 1D (red lines). Local exponents are calculated with a moving window shifted across the whole time series. Figures (A), (B) and (C) show the results for shifting window widths of 128, 256 and 512 s respectively. The heterogeneity of the rate of change in metabolic rate is revealed by the broad range of local scaling exponents $\alpha_{DFA}$, which shows a complex structure in time as opposed to the simpler and more restricted changes in the shuffled time series.

for this time series, we can see that observed $r(VO_2)$ fluctuations cannot be characterized by a single scaling exponent, and hence may be multifractal.

To determine whether this is the case, we examined whether the MF-DFA formalism can describe $VO_2$ fluctuations across different environmental temperatures. Figure 5 shows the average MF-DFA generalized fluctuation functions $F_q(s)$ calculated from time series measured at 30°, 20°, 10° and 0 °C (Figs. 5A, 5B, 5C and 5D respectively). Across all temperatures studied, and for all the values of $q$ examined, observed $F_q(s)$ functions show a crossover $\delta$ that defines two scaling regions, as shown by the fitted piecewise linear regressions (shown in black lines) (Fig. 5). Shuffling the time series leads to some changes in the crossover pattern, although no striking overall pattern may be discerned by qualitative

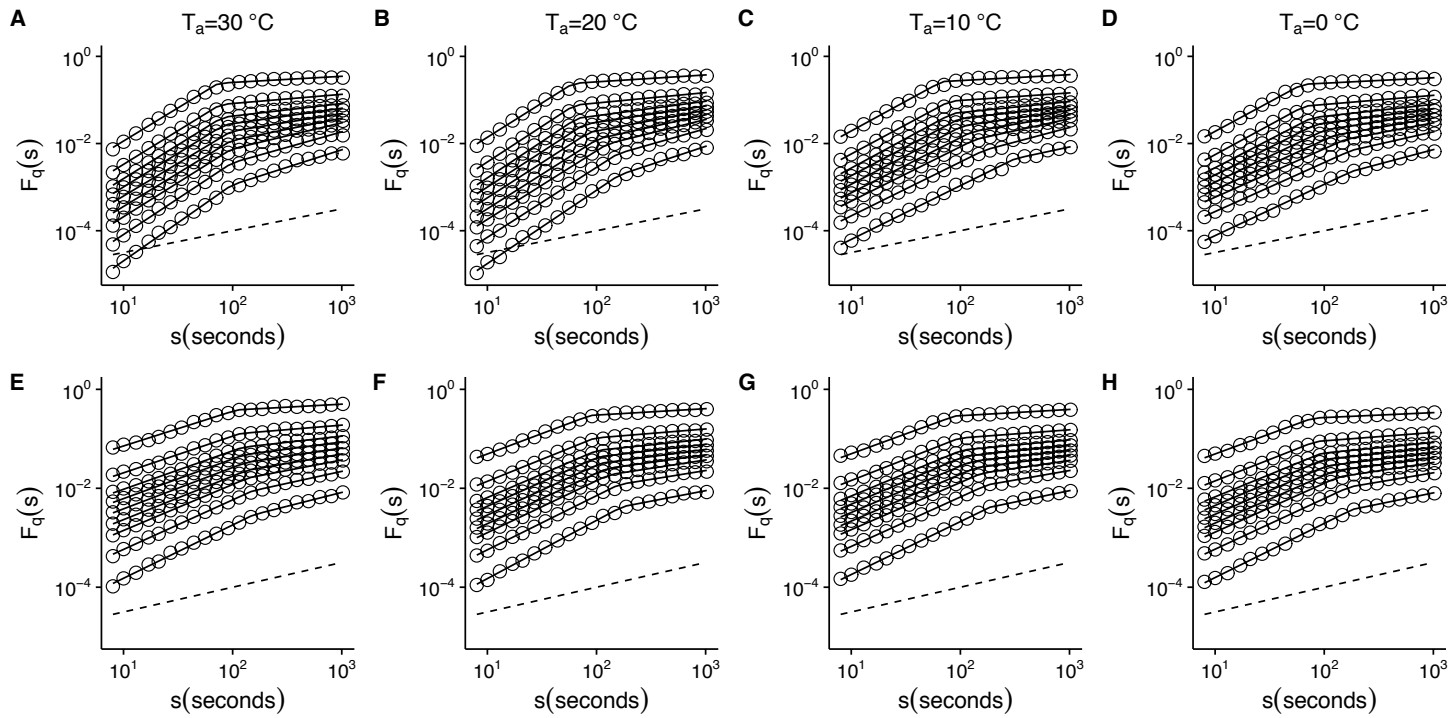

**Figure 5** **Temperature effects on generalized fluctuation function of $r(VO_2)$ in mice.** Figure shows log–log plots of the average generalized fluctuation function $F_q(s)$ as a function of time $s$ in $r(VO_2)$ time series. Figures (A)–(D) show the average $F_q(s)$ functions calculated for the observed $r(VO_2)$ time series measured at 30 °C, 20 °C, 10 °C and 0 °C respectively. Figures (E)–(F) show average $F_q(s)$ functions calculated for the AAFT shuffled $r(VO_2)$ time series measured at 30 °C, 20 °C, 10 °C and 0 °C respectively. Open circles in all figures show the observed $F_q(s)$ values for different values of $q$, with $q = 8, 4, 2, 1, 0, -1, -2, -4, -8$ (from the top to the bottom). Also shown in black lines are piecewise linear regression fits to the $F_q(s)$ functions. Dashed straight lines with slope $h = 0.5$ are shown below the data in each figure to allow qualitative comparison with the uncorrelated case. Please note that while only four experimental temperatures are shown, the remaining three temperatures show similar patterns.

examination (Fig. 5E–5F). It must be noted that while the remaining three series for 5°, 15° and 25 °C are not shown, they show similar patterns. In fact, detailed examination of the average generalized fluctuation functions reveals that $F_q(s)$ show the presence of crossover time scales $\delta$ for all temperatures studied, regardless of the order $o$ of the de-trending polynomial used (see Figs. S2–S8 for detailed results for different de-trending polynomial orders and all temperatures from 0 °C to 30 °C). Thus, for all temperatures examined, regardless of the order of de-trending polynomial used, we observed two scaling regimes are present, with the piecewise break point changing as a function of $q$ in some cases (see Fig. S9). While it could be argued that such scaling crossovers may be the result of trends associated with non-stationary dynamics in the data, examination of the Augmented Dickey-Fuller Test (ADF test) for all $r(VO_2)$ time series rejected the hypothesis of the presence of trends, and we observed that the ADF test yields $p < 0.01$ in all time series. Shuffling of the observed $r(VO_2)$ time series does not completely remove the crossover scales $\delta$ or the two observed regimes, but does seem to change the scaling exponent for the first scaling regime (see Figs. S2–S8). Given the presence of two scaling regimes across all time series studied, we then examined the scaling slopes of the curves for both of these scaling regimes and their change with the exponents $q$. This allowed us to estimate the average Hurst ($h(q)$) and Renyi ($\tau(q)$) spectra for each of these two scaling regimes. We

then also fitted the MMCM model to the observed Renyi ($\tau(q)$) spectra, and estimated the singularity spectra ($f(\alpha)$) based on these parameter fits.

When we examined average Hurst ($h(q)$) and Renyi ($\tau(q)$) spectra, as well as the corresponding singularity spectra ($f(\alpha)$) estimated from the MMCM fits on $\tau(q)$, we found that the two scaling regimes differ in their multifractal spectra across the seven temperatures studied. Figures 6A, 6D, 6G, 6J, 6M, 6P and 6S show the multifractality of $r(VO_2)$ fluctuations, as indicated by the dependence of $h(q)$ on $q$ for different temperature values. We find that fluctuations of different magnitudes in $r(VO_2)$ time series show different scaling behaviour, similar to what has been observed other complex systems (*Bunde & Lennartz, 2012*; *Kantelhardt et al., 2006*; *Kantelhardt et al., 2002*). However, the first and second scaling regimes differ in their behaviour, with smaller time scales (in the approximate range $8 \leq s \leq 100$) showing generalized Hurst exponent $h_1(q)$ values closer to 1.5, while larger time scales (in the approximate range $100 \leq s \leq 1,024$) show generalized Hurst exponents decreasing from $h_2(q) \approx 0.9$ to $h_2(q) \approx 0.25$ as the exponent order $q$ increases (Fig. 6). Hence, fluctuations on the first scaling regime show long-range correlations or persistence, similar to that of Brownian motion, regardless of the magnitude of the fluctuation. On the other hand, for the second scaling regime, small $VO_2$ fluctuations are characterized by larger scaling exponents $h_2(q)$, corresponding to power law, long-range correlated persistent dynamics, while larger $VO_2$ fluctuations present smaller $h_2(q)$ exponent values, corresponding to anti-persistent dynamics (see Figs. 6A, 6D, 6G, 6J, 6M, 6P and 6S). Thus, over intermediate to large time scales, large positive $r(VO_2)$ values are balanced by large negative values. On the other hand, for this range of scales, small $r(VO_2)$ values are persistent, such that small positive increases are followed by similarly valued changes, resulting in gradual positive trends in $VO_2$. A similar pattern occurs for negative rates of change, which leads to gradual negative trends in $VO_2$. Shuffling the $r(VO_2)$ time series results in markedly lower values of $h(q)$ scaling exponents for the first scaling regime, indicating the observed, persistent long-range correlation cannot be accounted for by a random sample of the observed spectral density function. On the other hand, in the second scaling regime, a complex response is observed, where shuffling results in changes only for negative and small positive $q$ values, whereas observed exponents for large positive $q$ values overlap with the exponents from shuffled time series. In fact, with the exception of 30 °C, very large fluctuations in *r(VO2)* do not differ from the random expectation (Fig. 6).

Observed differences in the range of $h(q)$ exponents for the two scaling regimes can also be observed when examining the Renyi exponent spectra. We observed mostly linear Renyi exponent spectra in the first scaling regime, while the second scaling regime shows nonlinear Renyi exponent spectra as expected for multifractal time series (*Kantelhardt, 2011*) (see Figs. 6B, 6E, 6H, 6K, 6N, 6Q and 6T). This suggests that the first scaling regime should either be monofractal or weakly multifractal, requiring a smaller range of scaling exponents to account for the observed singularities. On the other hand, the second scaling regime is characterized by strong multifractality, with a broader range of scaling exponent values. As observed in previous results, shuffling destroys the observed scaling spectra, with the exception of $\tau(q)$ values observed for positive $q$, which do not differ from the shuffled spectra (Fig. 6). In all the time series we examined, the observed Renyi exponent spectra

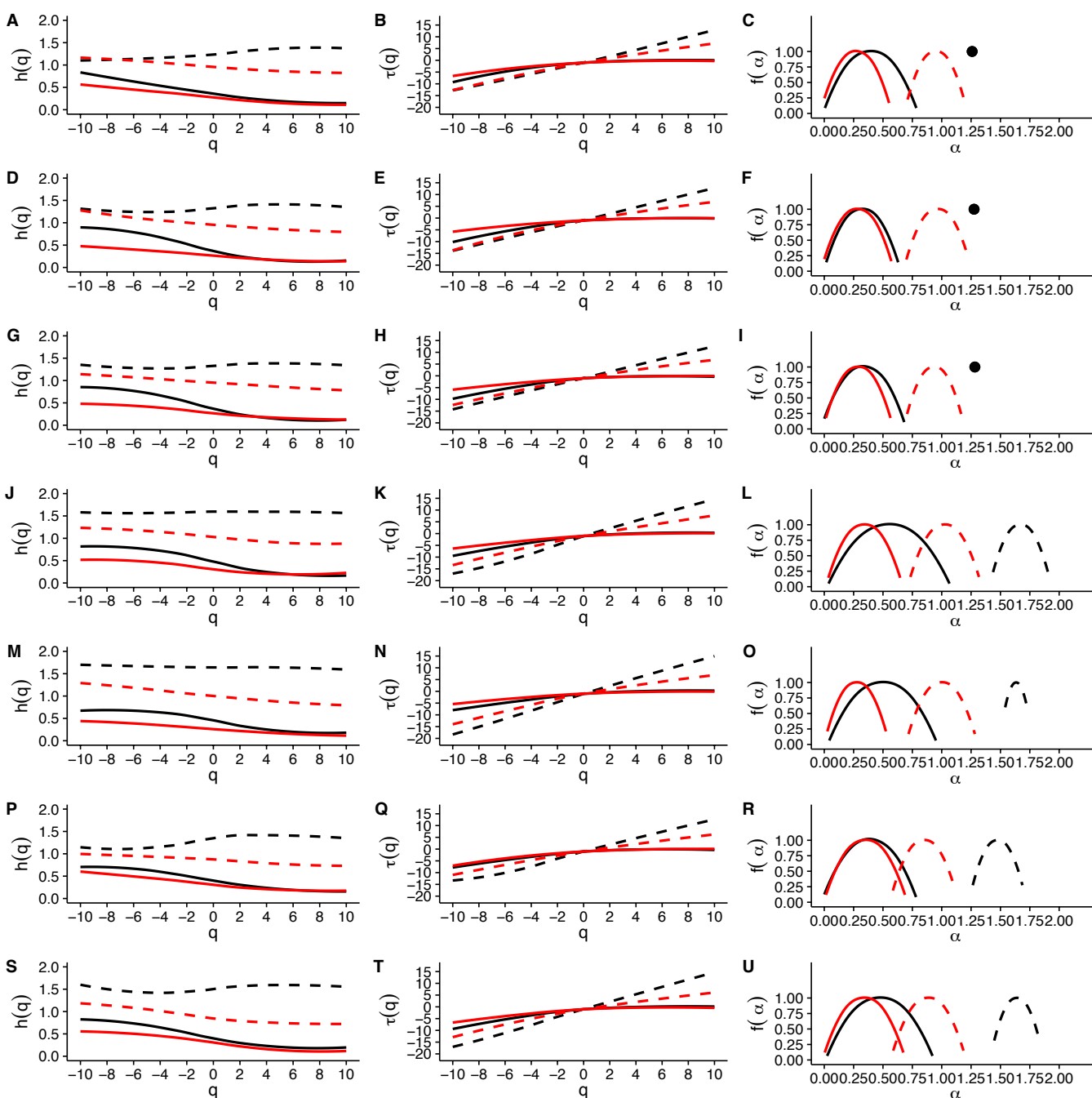

**Figure 6** **Multifractal Detrended Fluctuation Analysis of *Mus musculus r(VO2)* time series across different temperature treatments.** The figure shows the results of the multifractal scaling analysis for all mice studied. The results for the generalized Hurst exponent spectra ($h(q)$) are shown in figures (A), (D), (G), (J), (M), (P) and (S). Figures (B), (E), (H), (K), (N), (Q) and (T) show the results for the Renyi exponent spectra ($\tau(q)$). Figures (C), (F), (I), (L), (O), (R) and (U) show the results for the singularity spectra ($f(\alpha)$). Each figure shows in dashed and continuous black lines the smoothed conditional mean of the different spectra for the first and second scaling regimes respectively. For shuffled data, the smoothed conditional mean of the different spectra for the first and second scaling regimes are shown by dashed and continuous red lines respectively. For figures (C), (F) and (I), the singularity spectra of the first regime corresponds to a single point, shown by a filled circle. The singularity spectra reveal that for temperatures in the range $0\,^{\circ}\text{C} < T_a < 10\,^{\circ}\text{C}$ the time scales in the $8 < s < 100$ range present a monofractal scaling, while all remaining temperatures show a weak multifractal scaling. All data for the second scaling regime show strong multifractality, which is not completely lost when data are shuffled.

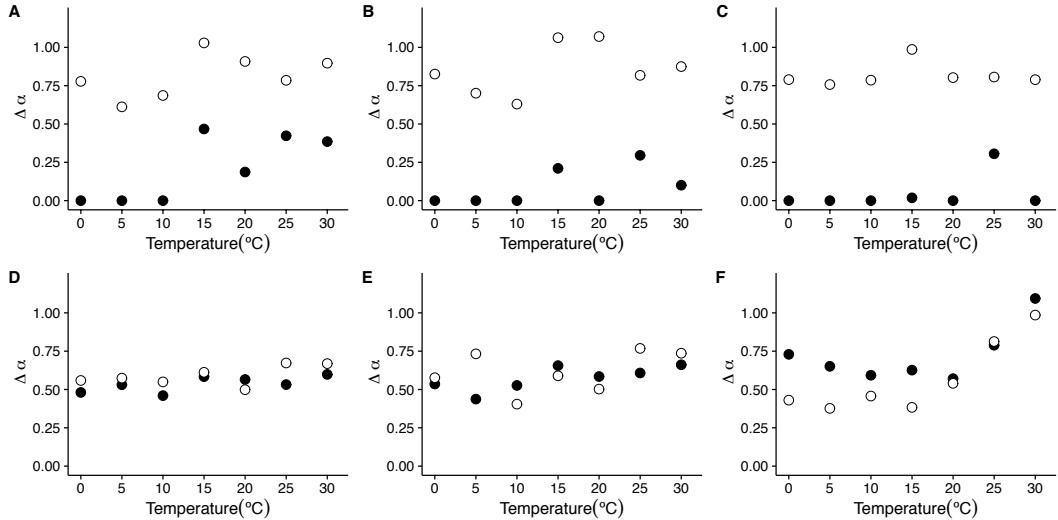

**Figure 7 Temperature effects on the strength of multifractality in mice.** The figure shows the average widths $\Delta\alpha$, of the $f(\alpha)$ spectra as a function of environmental temperature $T_a$. Figures (A)–(C) show the average widths $\Delta\alpha$ calculated for the raw $r(VO_2)$ time series with linear, quadratic and cubic polynomial detrending respectively. Figures (D)–(F) show the average widths $\Delta\alpha$ calculated for the AAFT shuffled time series with linear, quadratic and cubic polynomial detrending respectively.

were fit extremely well my the MMCM model shown in Eq. (8), with $R^2$ values for the nonlinear fitting procedure being close to 1.0 in all cases (see Fig. S10). This allowed us to use the fitted $\tau(q)$ values to estimate the singularity spectra $f(\alpha)$ for each individual, which were then averaged across all the different temperature treatments.

Examination of the average singularity spectra $f(\alpha)$ for different temperature treatments shows that the first scaling regime of these $r(VO2)$ time series are monofractal or weakly multifractal, as evidenced by either a single point or a narrower parabola in the $(\alpha, f(\alpha))$ plane (see dashed lines in graphs in Figs. 6C, 6F, 6I, 6L, 6O, 6R and 6U). These qualitative patterns do not change when quadratic or cubic de-trending polynomials are used (see right-hand columns of Figs. S11 and S12). Indeed, the average degree of multifractality, $\Delta\alpha$ shows that the first scaling regime the strength of multifractality decreases with temperature (see Fig. 7). While a similar qualitative pattern is observed for all de-trending polynomial orders, a the decrease with temperature is significant only for the linear de-trending case (linear OLS regression, $F = 8.202$, d.f. $= (1, 5)$, $p = 0.035$) (Figs. 7A, 7B and 7C). In sharp contrast, the second scaling regime shows broad singularity spectra, indicating a much larger degree of multifractality, $\Delta\alpha$ (see continuous lines in graphs on Figs. 6C, 6F, 6I, 6L, 6O, 6R and 6U). For this second scaling regime, no significant linear trends with temperature were observed, with the exception of the cubic de-trended data (linear OLS regression, $F = 13.43$, d.f. $= (1, 5)$, $p = 0.015$) (Fig. 7C). Shuffled data tend to show similar degrees of multifractality across different temperatures and orders of detrending polynomials (Figs. 7D–7F).

On the other hand, when we examine the exponent $\alpha_{max}$ of the singularity spectra, we see that the first scaling regime is characterized by much stronger singularities, with $\alpha_{max}$ taking

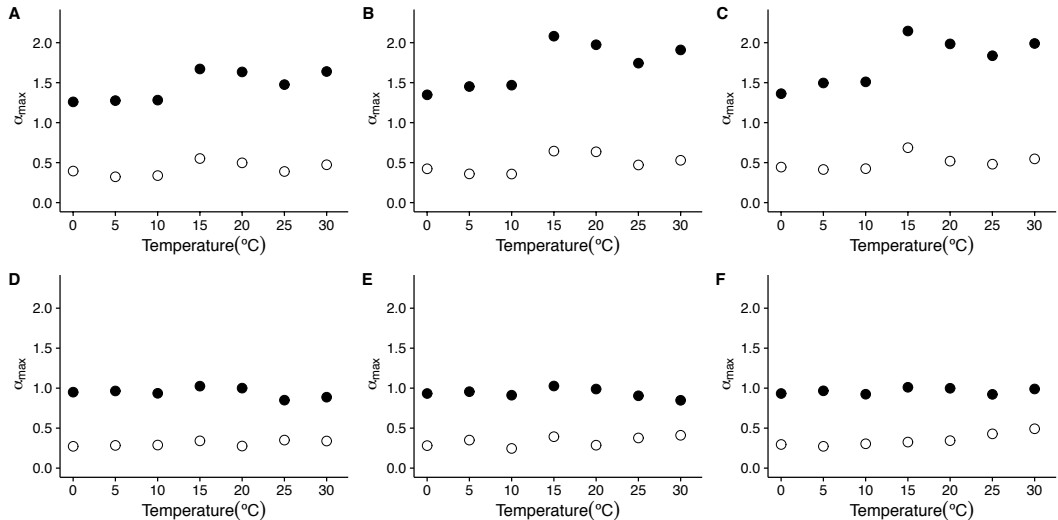

**Figure 8** **Temperature effects on the dominant multifractal exponent in mice.** The figure shows the average dominant fractal exponent $\alpha_{max}$, for the different the $f(\alpha)$ spectra as a function of environmental temperature $T_a$. Figures (A)–(C) show the average $\alpha_{max}$ values calculated for the raw $r(VO_2)$ time series using linear, quadratic and cubic polynomial detrending respectively. Figures (D)–(F) show the average $\alpha_{max}$ values calculated for the AAFT shuffled time series using linear, quadratic and cubic polynomial detrending respectively.

values closer to 1.5, being slightly larger for 15 °C and 20 °C (Figs. 6I and 6O). On the other hand, the second scaling regime is characterized by weaker stronger singularities, showing values of $\alpha_{max}$ below 0.5 (see Figs. 6C, 6F, 6I, 6L, 6O, 6R, 6U, 8C and 8F). Examination of the changes in $\alpha_{max}$ as a function of temperature for the first scaling regime indicates that the value of $\alpha_{max}$ has significant increases with temperature only for the linear and cubic cases (linear de-trending: $F = 7.52$, d.f. $= (1, 5)$, $p = 0.04$; cubic de-trending: $F = 7.52$, d.f. $= (1, 5)$, $p = 0.04$) (Figs. 8A–8C). In the case of quadratic de-trending, temperature values equal or greater than 15 °C show high values of $\alpha_{max}$, coherent with the persistent, Brownian motion-like values of $h(q)$ observed before. On the other hand, for the second scaling regime, $\alpha_{max}$ does not show significant changes with temperature for any de-trending order (Figs. 8A–8C). Shuffled data tend to show similar degrees of multifractality for different temperatures and orders of de trending polynomials, with shuffled data for the first scaling regime clustering around values close to $\alpha_{max} = 0.9$, and shuffled data for the second scaling regime clustering around values close to $\alpha_{max} = 0.3$ (Figs. 8D–8F). Thus, both the observed degree of multifractality $\Delta\alpha$, and the dominant multifractal singularity exponent $\alpha_{max}$ in these two scaling regimes cannot be attributed to random fluctuations.

## DISCUSSION

Physiological systems, and their state variables and signals, have been recognized as complex (*Burggren & Monticino, 2005*; *Glass, 2001*). To date, most studies examining the causes and functional implications of the loss of complexity in organisms have largely focused on human biomedicine, aiming to understand either pathologies or the senescence process (*Costa et al., 2008*; *Delignières & Torre, 2009*; *Goldberger et al., 2002*; *Hausdorff et al., 2001*;
*Lipsitz, 2004*). In this regard, our study aims to provide a better understanding of the role of physiological complexity in the homeostatic response to thermal challenges, particularly in the context of a changing world climate. Here, we analyzed the dynamics of metabolic rate fluctuations, $r(VO_2)$,under different *Ta's* using a well-studied model organism, the lab mouse *Mus musculus*. Using MF-DFA, our results show that within the *TNZ*, $r(VO_2)$ time series show two distinct scaling regimes in the fluctuation functions $F_q(s)$, with a crossover time scale $\delta$ of approximately $10^2$ s. Examination of the generalized *Hurst* exponents shows that these two scaling regimes correspond to persistent and anti-persistent dynamics for scales below and above the crossover time scale, with the strength of multifractality differing between these two regimes. When environmental temperature $T_a$ is decreased below the *TNZ*, the observed pattern of multifractal, anti-persistent long-range correlations over longer time scales does not vary a great deal. On the other hand, over short scales, the persistent long-range correlations transition from a weakly multifractal to a monofractal distribution. We now discuss these results.

The first aspect we discuss is the robustness of the rather complex long-correlation structure observed for our data. While previous analysis of $VO_2$ have reported long-range persistent $1/f^\beta$ fluctuations, described by a single dominant monofractal scaling exponent (*Chaui-Berlinck et al., 2002a*; *Chaui-Berlinck et al., 2002b*), we show here that that $VO_2$ fluctuations of different magnitudes are clustered throughout the experimental time series with varying types of long-range correlation, depending on the time scale analyzed. Thus, $r(VO_2)$ is a multifractal self-affine signal. This suggests that the feedback control mechanisms underlying rapid changes in energy consumption involve strongly non-linear dynamic processes. Both the observed multifractal exponent spectra and the scaling crossover differ from those observed under a random linear transformation in the frequency domain (*Kantelhardt, 2011*; *Schreiber & Schmitz, 1996*; *Schreiber & Schmitz, 2000*). This indicates that the observed multifractality of $r(VO_2)$ is a robust property of metabolic rate. The existence of this long-range correlation structure indicates the potential for plastic dynamic responses to thermal stress (*Goldberger et al., 2002*; *Ivanov et al., 2007*). In this regard, the existence of a crossover, with two characteristic long-range correlation signatures may be related to the dynamics of both $VO_2$ and $r(VO_2)$. As we have shown for data within the *TNZ* (see Fig. 1), $VO_2$ time series may show periods of higher energy consumption interspersed with periods of lower energy use (Fig. 1B). These periods present particularly different patterns of $VO_2$ changes, which are reflected in the pattern of $r(VO_2)$ fluctuations. Thus, higher average energy uses (larger mean $VO_2$ values) are associated with less variable values of $r(VO_2)$, in agreement with observed results for inter-specific scaling of $r(VO_2)$ across different vertebrate species (*Labra, Marquet & Bozinovic, 2007*), as well as in diverse complex systems (see references in *Labra, Marquet & Bozinovic, 2007*). Examination of $r(VO_2)$ data using different approaches Fourier power spectra, DFA and MF-DFA reveal that small-scale and larger scales present different scaling relationships. The first two methods agree qualitatively with the pattern shown by the MF-DFA $F_q(s)$ fluctuation functions. It is important point to out that that in all series, the scaling crossover was observed regardless of the de-trending polynomial order used in MF-DFA. On the other hand, the type of long-range correlation structure identified was

also robust. When data were analysed using MF-DFA using 2nd and 3rd order de-trending polynomials, the scaling regime for smaller time scales is observed to be either weakly multifractal or monofractal across most temperatures, while the second scaling regime is found to be multifractal for all three de-trending orders used in MF-DFA. For the second scaling regime, corresponding to larger time scales, the broadest singularity spectra are observed for 15 °C and 20 °C, with either $\alpha_{max} \approx 0.5$ for first de-trending order MF-DFA, or $0.5 > \alpha_{max} > 1.0$ for 2nd and 3rd de-trending order MF-DFA.

The second aspect we discuss is the possible explanations for the qualitative changes observed in the long-range correlation structure in the vicinity of 15 °C, as well as their potential significance. Metabolic rate changes are central for the control of $Tb$ in endotherms (*Chaui-Berlinck et al., 2005*; *Karasov & Martinez del Rio, 2007*). Thus, body temperature in these organisms is regulated through a complex set of processes and feedback relationships involving behavioral, endocrine, vasomotor and neural processes (*Chaui-Berlinck et al., 2005*; *Karasov & Martinez del Rio, 2007*). A recent review on the thermal physiology of *Mus musculus* shows that in this species the lower limit of normothermia ranges between 5 and 15 °C (*Gordon, 2012*). Below these temperatures, thermal homeostasis requires increased $VO_2$, which become nearly twice the BMR. These additional homeostatic requirements may be offset with different thermoregulation strategies that include behavioral, postural and physiological adjustments, all of which carry with them increased energetic costs. Over longer periods of time, these energetic requirements may not be met without resorting to alternative physiological strategies such as torpor (*Gordon, 2012*). Interestingly, individuals in our measurements did not reach the torpor stage, resorting only to individual huddling within the measurement chamber. Studies on thermoregulatory behavior have shown that small mammals such as lab mice form groups by huddling together as a behavioral thermoregulatory response to temperature challenges (*Canals, Rosenmann & Bozinovic, 1997*; *Canals et al., 1998*). Interestingly, this behavioral response behaves as a system with a continuous (second-order) phase transition, with a critical environmental temperature value found between 16 °C and 20 °C (*Canals & Bozinovic, 2011*). For low temperatures, individuals spontaneously aggregate, forming groups with a higher fractal dimension and a lower mass-specific metabolic rate. This change in behavior occurs in the same temperature range where we have observed maximal values for the degree of multifractality, supporting the idea that different physiological regimes may occur above and below this temperature range. Hence, future work could examine the long-range correlation properties of $VO_2$ fluctuations under different strategies such as torpor or group huddling, in order to determine whether the degree of multifractality decreases below that observed at 0 °C, giving rise either to monofractal scaling or to the loss of fractal autocorrelations.

A third point we discuss is the biological significance of these results. As mentioned earlier, whole-body metabolic rate is an emergent phenomenon, resulting from microscopic interactions with a large number of degrees of freedom and a complex set of opposing feedback mechanisms acting at different time scales (*Bozinovic, 1992*; *Chaui-Berlinck et al., 2005*). In this regard, the multifractal nature of metabolic rate highlights the complex and non-linear nature of the multiple feedback loops involved in the maintenance of physiological homeostasis (*Chaui-Berlinck et al., 2005*; *Darveau et al., 2002*; *Hochachka et*

*al., 2003*). The existence of multifractality in metabolic rate fluctuations has several interesting implications, particularly regarding the sensitivity to initial conditions. In general, multifractal dynamics are generated by non-linear recursive processes, which show different scaling or fractal properties depending on the initial conditions or on the particular history of external disturbances to the system (*Kantelhardt, 2011*). As a result, the observed singularities and scaling exponents of multifractal time series can change in time, leading to the presence of local abrupt shifts in the dynamics of these systems (*Kantelhardt, 2011*). In addition, these singularities are associated with the presence of both extreme events and fat tailed power law distributions, which have been shown to be a universal feature of metabolic rate across different vertebrate species (*Labra, Marquet & Bozinovic, 2007*). Despite the seemingly irregular unpredictable nature of metabolic rate fluctuations, our results show that they have a characteristic long-range correlation structure. Although in many applications the proximal mechanistic causes of observed fractality or multifractality have not been elucidated (*Kantelhardt, 2011*), the fact remains that multifractal processes such as $r(VO_2)$ are completely different from simple linear random fluctuations. This opens an interesting scenario regarding the potential use of multifractal properties as either a diagnostic tool or as baseline to determine animal response to environmental stress. This improved characterization may also eventually allow the modeling the dynamics and projection of the likelihood of extreme events or prediction of future behavior (*Kantelhardt, 2011*). This may complement the empirical estimates of metabolic rate, which typically correspond to the average value of $VO_2$ registered in a small section of the time series under specific environmental conditions (*Lighton, 2008*). Similarly, measurements of the rate of $VO_2$ under the maximum sustainable rate of exercise (i.e., maximal metabolic rate) have been shown to be mostly a function of aerobic capacity of the muscle mass (*Weibel et al., 2004*). In the light of our results, it seems reasonable to expect that $VO_2$ fluctuations under conditions of maximum sustainable exercise would also show multifractal long-term correlations as well as power law distributed fluctuations.

In addition to the physiological significance of long-range multifractal correlations of $r(VO_2)$, a related aspect pertains the taxonomic and systemic generality and significance of our results. It is relevant to discuss whether these observed patterns are expected to hold true for all endothermic species. While previous work on $r(VO_2)$ has reported a universal probability distribution function across different vertebrate species (*Labra, Marquet & Bozinovic, 2007*), no systematic comparative assessment has been carried out to determine if the long-range correlation structure may hold true for different endothermic species, be these birds or mammals. A particularly interesting aspect of such comparisons would be to examine the role of individual body size. Our work was carried out using a small endothermic species, the lab mouse. Analysis of a theoretical model of body temperature control by shifts in metabolic rate has suggested that the rate of heat loss and the capacity to rapidly increase metabolic output may lead to non-equilibrium between metabolic rate and body temperature in micro-endotherms (such as hummingbirds and small mice), resulting in non-random $1/f^\beta$ persistent oscillations of $VO_2$, even within the *TNZ* (*Chaui-Berlinck et al., 2002a*). Our results indicate that $VO_2$ are not only long-range correlated, but that have a complex multifractal structure, which indicates that the model of *Chaui-Berlinck*

 

*et al. (2002a)* yields predictions that are at least qualitatively correct. Interestingly, this theoretical model also predicts that larger endotherms such as the rat may not exhibit similar complex oscillations, due to a dynamic equilibrium between metabolic rate and body temperature, given the smaller surface area-volume ratio. If correct, this model predicts the absence of long-range correlated $r(VO_2)$ oscillations for larger endotherms, with multifractal dynamics being found only in micro-endotherms, regardless of whether they are mammals or birds. Whether a threshold body size may be identified below which multifractality may be observed would indicate the onset of a highly nonlinear configuration of control processes acting in the regulation of body temperature. The alternative outcome would be that multifractal long-range correlations also hold true for larger endotherms. This alternative scenario would indicate that a more detailed model analysis is required to account for the processes affecting metabolic rate oscillations.

## GENERAL CONCLUSION

While an increasing number of authors have pointed out the complex nature of physiological processes (*Burggren & Monticino, 2005*; *Spicer & Gaston, 1999*), an emerging research question is what are the consequences and implications of physiological complexity for the homeostatic adaptive capability of animals, particularly on a scenario of global climate change. In addition to considering the potential role of organism body size, it is important to determine whether the observed multifractal correlation structure is a general trait of all endotherm taxa, or if it is a characteristic trait of mammals as a lineage. Comparative experimental studies may help to untangle the relative importance of body size and taxonomic inertia in the emergence of multifractality. A related question is whether ectotherms do present any long-range correlation structure in their metabolic rate dynamics. If complexity is an emergent characteristic arising from the different thermal control feedback loops, then multifractality should be absent in metabolic rate dynamics of reptiles or amphibians. The goal of such studies would be to allow the assessment of the relative importance of universal emergent statistical behaviour and phylogenetic inertia in morphological and physiological traits that may give rise to complex metabolic rate fluctuations. Again, the use of a comparative, controlled experimental approach may allow careful examination of the relationships between the complexity of metabolic rate dynamics and the origins of endothermy.

Our results show that the dynamic response of the metabolic machinery in a model mammal species facing thermal challenge do not reduce themselves to the linear variance response expected, evidencing in addition that this response is regulated by environmental history experienced of individual. In this regards, the humped shape observed from the relationship between complexity level of $VO_2$ and decrease of temperature agree with a limit at the physiological capability to control of body temperature. Future work in this area may focus on experimental explorations of the physiological basis of long-term correlations and multifractality of $VO_2$ fluctuations. For example, such work may examine the relative importance of different control mechanisms regulating the rate of oxygen uptake as part of a hierarchical cascade of feedback loops that lead to multifractality.

## ACKNOWLEDGEMENTS

We thank F Boher and S Clavijo for their assistance during the development of these experiments. FAL thanks C Huerta and E Labra for their continued support.

### Funding

This work was supported by grants from Fondo Nacional de Ciencia y Tecnología 1100729 to FAL and Fondo Nacional de Ciencia y Tecnología 1130015 and Fondo Basal FB-0002 to FB. The funders had no role in study design, data collection and analysis, decision to publish, or preparation of the manuscript.

### Grant Disclosures

The following grant information was disclosed by the authors:
Fondo Nacional de Ciencia y Tecnología: 1100729 and 1130015.
Fondo Basal: FB-0002.

### Competing Interests

The authors declare there are no competing interests.

### Author Contributions

- Fabio A. Labra conceived and designed the experiments, analyzed the data, wrote the paper, prepared figures and/or tables, reviewed drafts of the paper.
- Jose M. Bogdanovich conceived and designed the experiments, performed the experiments, analyzed the data, contributed reagents/materials/analysis tools, wrote the paper, reviewed drafts of the paper.
- Francisco Bozinovic conceived and designed the experiments, performed the experiments, contributed reagents/materials/analysis tools, wrote the paper, reviewed drafts of the paper.

### Animal Ethics

The following information was supplied relating to ethical approvals (i.e., approving body and any reference numbers):

Care of experimental animals was in accordance with institutional guidelines, and experimental protocols followed were approved by the following Review Boards:

Bioethics committee, Universidad Santo Tomás.

Bioethics committee, Pontificia Universidad Católica de Chile.

Bioethics committe, Chilean National Committee of Science and Technology (CONICYT).

All three review boards issued an approval letter, indicating the endorsement of animal care and experimental protocols.

### Data Availability

The raw data has been supplied as a Supplementary File.

## Supplemental Information

Supplemental information for this article can be found online at http://dx.doi.org/10.7717/peerj.2607#supplemental-information.

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
