# Peer review of "Nonlinear temperature effects on multifractal complexity of metabolic rate of mice"

_PeerJ, doi:10.7717/peerj.2607_

## Round 0.1 · original submission · Major Revisions

Dear Dr. Fabio A Labra,

I have received the review reports on your manuscript, "Nonlinear temperature effects on multifractal complexity of metabolic rate", which you submitted to PeerJ. Based on the advice received, your manuscript could be reconsidered for publication should you be prepared to incorporate major revisions.

Reviewer 1 ·

Basic reporting

No Comments

Experimental design

No Comments

Validity of the findings

No Comments

Additional comments

1. Many physical parameters, such as fluctuation function, singularity spectra, long-term correlation structure, et al., lack important details about the analysis. What are physical meanings of the results, especially for oxygen consumption time series?
2. In the Fig.1, fluctuation function is found to be a broken power law. For the larger time scales (scales>100 sec), the scaling exponent <0.5. In line 263-265, “…, with the exponent for large time scales being close to 1, and with a much larger value for smaller time scales”. The fig.1 and the statements are inconsistent.
3. In the Fig.1, For the larger time scales (scales>100 sec), the scaling exponent <0.5. This is the case of anti-persistent. What are physical meanings of the results, especially for oxygen consumption time series?
4. In line 292-293, why choose the large times scales (100<s<2500)? Why not choose the scales<100 sec?
5. In line 290, “…, they show similar patterns.” What you mean about “similar patterns”?

Reviewer 2 ·

Basic reporting

No Comments

Experimental design

No Comments

Validity of the findings

No Comments

Additional comments

With necessary revision, this paper may be considered for possible publication. Major issues that need to be clarified include:
(1) Eqs.2 and 3 are so blurred that I cannot understand very well its meaning. Please rewritten Eqs.2 and 3 in resubmitted version.
(2) Since the authors thought that the metabolic rate is nonlinear, so DFA analysis seems to be unnecessary. MF-DFA is an extension to DFA and is completely used for validation of long range correlation and multifractal characteristics of nonlinear time series.
(3) In Fig.3, local exponents are calculated with a moving window of 512 seconds across the whole time series. Why to choose a moving window of 512 seconds? What is the selection criteria?
(4) In Fig.4, what does the MF-DFA2 mean? The concept was not found in the article. Please give a specific explanation.
(5) Axis titles are not unified, such as temperature (℃) in Fig.7.

Reviewer 3 ·

Basic reporting

Clear and professional English language uese throughout.
Literature well referenced and relevant.
Figures are relevant and well labelled.

Experimental design

Research question well defined, relevant and meaning.
Methods described with sufficient detail to replicate.

Validity of the findings

Conclusion well stated, linked to original research question.

Reviewer 4 ·

Basic reporting

see general comments to the authors

Experimental design

see general comments to the authors

Validity of the findings

see general comments to the authors

Additional comments

I do not think that the work presented in this paper deserves publication in PeerJ in its present form. Although the issue addressed in this manuscript, namely a
multifractal analysis of metabolic rate time series towards a better understanding of the role of environmental temperature on the metabolic machinery, I have three kinds of comments and objections. The first ones are of conceptual and fundamental grounds; the second ones concern technical issues; the third ones result from some
skepticism about the relevance of the reported results.

Theoretical background
I am afraid that the authors have not fully understood the deep analogy between the multifractal formalism and statistical thermodynamics. In the Introduction there is some confusion concerning the concepts of long-range correlations (LRC), power-law scaling of power spectra or partition functions, monofractal or multifractal
distributions. Many misleading statements are made due to some oscillations between two different kinds of mathematical objetcs : (i) measures, noises (e.g.
r(VO2)=log10(VO2(t+1)/VO2(t)) that are not continuous distributions and (ii) functions (e.g. the cumulated z(t)=Σ r(VO2)) that are continuous distributions. For example the sentence :
« Multifractal signals reveal a stronger degree of nonlinearity, and may be characterized as statistically inhomogeneous, in attention to the fact that different sections of the signal have different scaling properties » is totally misleading. Such
signals are indeed piece-wise monofractal signals. A multifractal function has a Hölder regularity that fluctuates from point to point, the local stregth h of a singularity resulting from the local accumulation of all the other singularities. This means that if by
some abstract operation one isolates all the singularities of given Hölder exponent h* removing all the other ones from the data, then the new data will not be monofractal with a unique singularity strength h* but multifractal with a singularity spectrum D(h) which is maximum at h*.
The multifractal formalism has been originally introduced in the early eighties (box-counting algorithms) to characterize the complexity of measures (specially invariant measures of nonlinear dynamical systems). It has been generalized to functions in the late eighties (early nineties), first by using the structure function method and then improved by using the so-called wavelet transform modulus maxima (WTMM) method. We refer the authors to the following original article :

Multifractal formalism for fractal signals: the structure function approach versus the wavelet transform modulus maxima method.
J.F. MUZY, E. BACRY et A. ARNEODO, Phys. Rev. E 47, 875 (1993)

and to the review articles :

The multifractal formalism revisited with wavelets.
J.F. MUZY, E. BACRY et A. ARNEODO, Int. J. of Bifurcation and Chaos 4, 245 (1994)

The thermodynamics of fractals revisited with wavelets.
A. ARNEODO, E. BACRY et J.F. MUZY, Physica 213 A, 232 (1995)

Later the detrended fluctuation analysis (DFA) has been phenomenologically
introduced by E. Stanley’s group. As mentioned by the authors, DFA has the advantage to be easy to implement. Unfortunately this method has the disadvantage to be used without having deeply understood the theoretical concepts underlying the multifractal formalism. We refer the authors to the following paper where the DFA and WTMM methods are compared :

Wavelet-based estimators of scaling behavior.
B. AUDIT, E. BACRY, J.F. MUZY and A. ARNEODO, IEEE Transaction Information Theory 48, 2938 (2002)

Suggestion : Since the authors actually perform DFA analysis on the cumulated z(t)=Σ r(VO2) function, I suggest that from the first sentence of the introduction, the authors discuss the concepts of LRC, scaling, monofractal, multifractal in the context of multifractal analysis of functions.
Thus for monofractal functions (e.g. fBm of index H) : the Fourier power spectrum decays as a power law with exponent β=2H+1, where (i) H=1/2 corresponds to functions with uncorrelated increments, (ii) H>1/2 LRC increments and (iii) H<1/2, anti-correlated increments; τ(q)=qH-1 and D(h) reduces toa single point h=H.
For multifractal functions : the Fourier power spectrum decays as a power law with exponent β=2h(q=2)+1; τ(q) is nonlinear and D(h) has the shape of a bump with hmax=h(q=0) (and not h(q=2) as given by the power spectrum exponent).

Methodology
The authors use the DFA methodology but they do not seem to have much expertise in performing multifractal analysis. The application of this methodology requires some cautions in order to avoid to be biased by finite-size effects and lack of statistical convergence. As far as power spectrum analysis is concerned, the authors have compared their results for r(VO2) to those of a shuffled time-series. They have shown that LRC behavior is observed at high frequency (Fig. 1e) meaning at small time and then they have performed DFA analysis on cumulated r(VO2) at large time without comparing with results obtained for shuffled time-series (Fig. 4) ????????? There is clearly a time scale around 100 s that separates two different scaling regimes (Figs 2 and 4). This time scale does not seem to depend on q; this means that it is a physical time scale separating two distinct scaling regimes that require to be analyzed
separately (and not the evidence of a phase transition in the singularity spectrum). The range of q values investigated from -8 to 8 looks quite large and is probably misleading. As performed in Fig. 2, the robustness of the results of DFA analysis in Figs 4 to 7 have to be tested by investigating different orders of detrending polynomials.

Results
As presented in this original manuscript, the results and conclusions are totally unreliable. Besides the issues and suggestions previously mentioned in this report, the authors have to address the following points :
(i) discuss and interpret the observed characteristic time scale around 100s that clearly separates two different scaling regimes at small and large times;
(ii) perform a comparative DFA analysis of cumulated r(VO2) and cumulated shuffled time-series over the two scaling time ranges;
(iii) compare for both regimes the multifractal spectra obtained at different temperatures;
(iv) discuss what is learnt from a metabolic point of view from the analysis of these two regimes.

In the context of multifractal analysis of biological and physiological signals, I bring to the attention of the authors the following works on the application of the WTMM method to DNA sequences and to infrared thermograms for breast cancer diagnosis :

Multi-scale coding of genomic information : from DNA sequence to genome structure and function.
A. ARNEODO, C. VAILLANT, B. AUDIT, F. ARGOUL, Y. D’AUBENTON-CARAFA & C. THERMES, Physics Report 498, 45 (2011)

Wavelet-based multifractal analysis of dynamic infrared thermograms to assist in early breast cancer diagnosis.
E. GERASIMOVA, B. AUDIT, S.G. ROUX, A. KHALIL, O. GILEVA, F. ARGOUL, O. NAIMARK & A. ARNEODO, Front. Physiol. 5, 176 (2014)

In conclusion, I am not convinced by the relevance of the results reported in this manuscript mainly because of the lack of care and rigor in performing the DFA analysis. The authors should have trained on synthetic monofractal (fBms) and multifractal time-series prior to application to experimental data. I therefore suggest that the manuscript be returned to the authors and that it be deeply revised taking
into account the above comments and suggestions.

Annotated reviews are not available for download in order to protect the identity of reviewers who chose to remain anonymous.

---

## Round 0.2 · accepted · Accept

Dear authors,

After reading your revised manuscript and the review reports from anonymous reviewers, I decide to accept your paper to be published in PeerJ.

Reviewer 1 ·

Basic reporting

No Comments

Experimental design

No Comments

Validity of the findings

No Comments

Additional comments

The sentence structure and content in the revised paper is much better than in the first version of the paper. If the paper structure and length meet the PeerJ standard, this paper may be considered for possible publication.

Reviewer 2 ·

Basic reporting

No Comments

Experimental design

No Comments

Validity of the findings

No Comments

Additional comments

It is my opinion that with the suggested revisions to all the concerns raised, the publication will be a good contribution to the journal and the scientific community interested in this problem. Additionally, I would like to thank the authors for investing your time and effort to answer my questions and address my concerns in this review process.